# Alpha7 Nicotinic Acetylcholine Receptor Antagonists Prevent Meningitic *Escherichia coli*-Induced Blood–Brain Barrier Disruptions by Targeting the CISH/JAK2/STAT5b Axis

**DOI:** 10.3390/biomedicines10102358

**Published:** 2022-09-22

**Authors:** Zelong Gong, Xuefeng Gao, Yubin Li, Jinhu Zou, Jingxian Lun, Jie Chen, Chengxing Zhou, Xiaolong He, Hong Cao

**Affiliations:** Department of Microbiology, School of Public Health, Southern Medical University (Guangdong Provincial Key Laboratory of Tropical Disease Research), Guangzhou 510515, China

**Keywords:** bacterial sepsis and meningitis, alpha7 nicotinic acetylcholine receptor, blood–brain barriers, cytokine-inducible SH2-containing protein

## Abstract

Despite the availability of antibiotics over the last several decades, excessive antibiotic treatments for bacterial sepsis and meningitis (BSM) in children may result in several adverse outcomes. Hematogenous pathogens may directly induce permeability increases in human brain microvascular endothelial cells (HBMECs) and blood–brain barrier (BBB) dysfunctions. Our preliminary studies demonstrated that the alpha7 nicotinic acetylcholine receptor (α7nAChR) played an important role in the pathogenesis of BSM, accompanied by increasing cytokine-inducible SH2-containing protein (CISH) at the transcriptome level, but it has remained unclear how α7nAChR-CISH works mechanistically. The study aims to explore the underlying mechanism of α7nAChR and CISH during *E. coli*-induced BSM in vitro (HBMECs) and in vivo (α7nAChR-KO mouse). We found that in the stage of *E. coli* K1-induced BBB disruptions, α7nAChR functioned as the key regulator that affects the integrity of HBMECs by activating the JAK2–STAT5 signaling pathway, while CISH inhibited JAK2–STAT5 activation and exhibited protective effects against *E. coli* infection. Notably, we first validated that the expression of CISH could be regulated by α7nAChR in HBMECs. In addition, we determined the protective effects of MLA (methyllycaconitine citrate) and MEM (memantine hydrochloride) (functioning as α7nAChR antagonists) on infected HBMECs and suggested that the α7nAChR–CISH axis could explain the protective effects of the two small-molecule compounds on *E. coli*-induced HBMECs injuries and BBB disruptions. In conclusion, we dissected the α7nAChR/CISH/JAK2/STAT5 axis as critical for the pathogenesis of *E. coli*-induced brain microvascular leakage and BBB disruptions and provided novel evidence for the development of α7nAChR antagonists in the prevention of pediatric *E. coli* BSM.

## 1. Introduction

Despite the availability of antibiotics over the last several decades, an unmet requirement still exists for new practical therapeutic approaches to treat pediatric infections, especially in developing countries [1,2]. Most pediatric infection-induced deaths were caused by bacterial sepsis and meningitis (BSM), which is characterized by high mortality and adverse prognosis [3]. Recently, CHINET (China bacterial resistance monitoring network) reported that “as the most common gram-negative bacterial pathogens causing BSM, Escherichia coli (*E. coli*) exhibits an uncontrollable, increasing multi-antibiotics resistance” (http://www.chinets.com, accessed on 25 May 2022). Antibiotics are effective in treating pediatric BSM, but antibiotic overuse increases the risk of drug resistance [3,4]. In addition, excessive antibiotic treatments in BSM patients were related to several adverse outcomes, including necrotizing enterocolitis (NEC), late-onset sepsis (LOS), and even death [5,6]. Recent research has shown that conventional antibiotic treatment for an infection induced by gram-negative bacteria commonly leads to severe inflammation associated with LPS, accompanied by the production of a large number of pro-inflammatory factors [7]. Hence, the crosstalk between host–pathogen interactions has become a new strategy and research point for developing antibiotic replacement therapy.

Chemokines are involved in infection-related inflammation and induce permeability increases in human brain microvascular endothelial cells (HBMECs), which are a significant indicator for evaluating BBB integrity [8,9]. Our preliminary studies demonstrated that alpha7 nicotinic acetylcholine receptor (α7nAChR) played a detrimental role in the pathogenesis of BSM and BBB disruptions, accompanied by an increase in cytokine-inducible SH2-containing protein (CISH) at the transcriptome level [10,11]. Recent studies suggested that the activation of Janus kinase signal transducer and transcriptional activator (JAK/STAT) were positively correlated with BBB disruptions [12], while JAK2 phosphorylation depends on activation of α7nAChR [13]. Whether these biological signaling pathways (mediated by α7nAChR and CISH) could explain the pathogenesis of BSM is still unknown. Are there any more complicated interactions that exist?

This study explored the underlying mechanism of α7nAChR, CISH, and their downstream signaling pathways during pediatric *E. coli*-induced BSM. By involving in vitro (human brain microvascular endothelial cells infected with *E. coli*) and in vivo (wild-type and α7nAChR knockout mice intragastric administration with *E. coli*) models, we evaluated the α7nAChR/CISH/JAK2/STAT5 axis in the pathogenesis of *E. coli*-induced BBB injuries and brain microvascular leakage. In addition, methyl lycaconitine citrate (MLA) and memantine hydrochloride (MEM) were two small-molecule compounds targeting α7nAChR, used during *E. coli* K1 infection attenuated BBB injuries.

## 2. Materials and Methods

### 2.1. Bacterial Strains

#### Bacterial Strains, Isolation, and Culture

*Escherichia coli* K1(E44) is derived from the RS218 strain (O18: K1: H7), which was isolated from a cerebrospinal fluid (CSF) sample of a neonatal BSM patient and showed rifampicin-resistant properties [14,15]. Bacteria were cultured and stored in brain heart infusion (BHI) broth with 20% glycerol at −80 °C.

### 2.2. Cell Assays In Vitro

#### 2.2.1. Cell Cultures and Reagents

The immortalized human brain microvascular endothelial cells (HBMECs) were isolated and cultured as described previously [16,17]. HBMECs were cultured in a 5% CO_2_ humidified incubator at 37 °C in 90% Dulbecco’s modified Eagle’s medium (DMEM) containing 10% fetal bovine serum (FBS) (Gibco, Gaithersburg, MD, USA) and 1% nonessential amino acids (NEAAs) (Sigma, St. Louis, MO, USA). Phosphate buffer saline (PBS, pH = 7.4) was obtained from Shanghai Double Spiral Biotech Co., Ltd. (Shanghai, China).

#### 2.2.2. siRNA and Plasmid Transfection

HBMECs in 6-well plates (1 × 10^5^ cells/well) were cotransfected with GV492-CISH plasmid (Genechem, Shanghai, China) (more specific information is provided in supplementary) and siRNA (RIBOBIO, Guangzhou, China) (including siRNA-1: CCACCAATGTACGCATTGA; siRNA-2: CCACTGCTGTACACCTAAA; siRNA-3: GACACACACCTGCAGAAGAT) using Lipofectamine 3000 reagent (Invitrogen, Waltham, MA, USA) or riboFECT^TM^ CP Transfection (RIBOBIO). The vector GV492-CISH (MW: 12.7 kb) was identified by restriction enzyme digestion (*EcoRI*, *XhoI*) and nucleotide sequencing. Then, the cells were incubated for 2–3 days, and the downregulation or upregulation of CISH protein expression in the transfected HBMECs was detected by Western blotting.

#### 2.2.3. Cell Viability Assays

The in vitro evaluation of cell viability was performed by a cell counting kit-8 (CCK-8) (Dojindo, Kumamoto, Japan). HBMECs were inoculated in 96-well plates (excluding the edge effect) and divided into a control group, *E. coli* E44 infection group, *E. coli* E44+MLA pretreatment group, and *E. coli* E44+MEM pretreatment group. After pretreatment for 2 h, sterile normal saline and *E. coli* E44 were added to the control group and *E. coli*. E44 infection for 2 h, respectively. Then, CCK-8 (10 μL/well) reagents were added to each well and incubated at 37 °C for 1–2 h. Then, the absorbance was measured at 450 nm by an Infinite M200PRO (TECAN, Männedorf, Switzerland).

#### 2.2.4. Adhesion and Invasion Assays

Adhesion and invasion assays on *E. coli* E44 were performed following a standard method [17]. Adhesion assay: *E. coli* E44 were prepared at a concentration of 10^7^ CFU/mL. HBMECs were infected with *E. coli* E44 for 2 h with a multiplicity of infection (MOI) = 100 at 37 °C in 5% CO_2_. Then, the infective HBMECs were washed three times with PBS solution. The adhesive abilities of *E. coli* E44 in different groups were measured as the number of bacteria adhering to the host cell. Invasion assay: As mentioned above, HBMECs were infected with *E. coli* E44 for 2 h, washed three times with PBS solution, and covered with DMEM containing gentamicin (200 μg/mL) to kill extracellular bacteria. Afterwards, the invasive abilities of *E. coli* E44 in different groups were measured as the number of bacteria inside the body of HBMECs.

#### 2.2.5. Western Blotting

The Western blotting protocol follows a previous work [18]. Infective HBMECs were collected with a mixture of phenylmethylsulfonyl fluoride (PMSF) (Besbio, Shanghai, China), radioimmunoprecipitation assay (RIPA) (Besbio), and phosphatase inhibitors (Glpbio, Montclair, CA, USA). PVDF membranes were blocked with skim milk (5%) in TBST for 1 h at room temperature and incubated with a rabbit anti-mouse polyclonal JAK2, p-JAK2 antibody (diluted 1:3000) (Abcam or Proteintech, Rosamond, CA, USA); STAT5b, p-STAT5b and occludin antibody (diluted 1:2000) (Proteintech, Rosamond, IL, USA); and CISH (diluted 1:1000) (Abclone Technology, Wuhan, China) antibody at 4 °C overnight. The membranes were reacted with a 1:1000 dilution of horseradish peroxidase (HRP) goat anti-rabbit IgG (Dingguo, Beijing, China) for 1 h at room temperature. After being washed three times in TBST, the membranes were reacted with a mixture of peroxide solution (Bio-Rad, Hercules, CA, USA) and luminol enhancer solution (Bio-Rad) (1:1). Finally, the WB results were detected by a Tanon 5500 chemiluminescent imager system (Tianneng, Huzhou, China), referencing β-actin protein for standardization.

#### 2.2.6. Cell Immunofluorescence

HBMECs were fixed with 4% paraformaldehyde for 30 min at room temperature. Then, 0.5% Triton X-100 was added to the cell slide for 20 min for cell transparency. After that, the cells were washed with PBST and blocked with bull serum albumin (BSA) (Abcone, Shanghai, China) for 1 h. Finally, the cells were incubated overnight with anti-CISH antibody (Rabbit) at 4 °C. Subsequently, after washing 3 times, the cells were covered with goat anti-rabbit IgG (H&L)-FITC and goat anti-rabbit IgG (H&L)-TRITC fluorescent secondary antibodies for 1 h. For the purpose of visualizing cell nuclei, HBMECs were mixed with DAPI in the dark for 20 min. Finally, the cell slides were observed under an optical microscope (with Axio Imager M2 software 4.50) (Nikon, Tokyo, Japan) (with ×200, ×400 magnification) [19].

### 2.3. Animal Assays In Vivo

#### 2.3.1. Ethics Statement

Animal experiments were approved by the Animal Care Committee of Southern Medical University (Guangzhou, China) (with protocol number No. 2018019, 09/05/2018). One- to two-week-old C57 mice were obtained from the Animal Experimental Center of Southern Medical University. a7nAChR heterozygous (α7nAChR+/−) mice with a C57BL/6J background were purchased from the Jackson Laboratory (B6.129S7-Chrna7tm1Bay/J). Littermate α7nAChR−/− and α7nAChR+/+ (wild-type) mice were generated from heterozygous mice for the experiments.

#### 2.3.2. Animals and Experimental Design

α7nAChR-KO mice (C57BL/6 background) were purchased from The Jackson Laboratory (B6.129S7-Chrna7tm1Bay, number 003232). In the α7nAChR-KO mouse assay, 12 mice (wild type: *n* = 6; α7nACR-KO: *n* = 6) were used in the experiments. PCR methods and agarose gel electrophoresis (α7nAChR- primer 1: CCCTTTATAGATTCGCCCTTG; α7nAChR- primer 2: ATCAGATGTTGCTGGCATGA; α7nAChR- primer 3: TTCCTGGTCCTGCTGTGTTA) were applied for genetic identification, the band of α7nACR-KO appeared at 189 bp.α-bungarotoxin (Biotium, Fremont, CA, USA) assay was used to detect the binding activation of α7nAChR. In the MLA and MEM assays, thirty-two mice were randomly divided into four groups (control group, *E. coli* E44 group, *E. coli* E44+MLA, and *E. coli* E44 + MEM group). Three days before infection, mice were given sterile normal saline, MLA (5.0 mg/kg), or MEM (20.0 mg/kg) by intraperitoneal injection, respectively. After drug pretreatments, mice in different groups were infected with *E. coli* E44 (1 × 10^7^ CFU/mouse) by intraperitoneal administration.

#### 2.3.3. Bacterial Load (BACT) in Mice Peripheral Blood and Cerebrospinal Fluid

After infection for 8 h, mice in different groups were sacrificed to collect peripheral blood, cerebrospinal fluid, and brain tissues. To determine the pathogenesis of BSM, BACT in the blood and the cerebrospinal fluid of mice was detected by the dilution plate counting method. Diluted blood and cerebrospinal fluid were incubated on plates containing rifampicin. After culturing at 37 °C for 18 h, the counts of bacterial colonies were recorded for analysis.

#### 2.3.4. Hematoxylin-Eosin and Immunohistochemistry

The whole brains of mice were collected and fixed in 10% formalin buffer solution for 24 h. Sample pretreatment (dehydration, embedding, sectioning, hyalinization, H&E staining, and sealing) was performed according to a previous method [17]. To determine JAK2 activation in brain tissues, a standardized method was adopted [18]. Anti-JAK2 polyclonal antibody (Proteintech, Rosemont, IL, USA) was mixed 1:500 in 5% skim milk. The sections were covered and incubated with polyclonal solutions (100 μL) at 4 °C for approximately 24 h. Then, the slides were washed with TBST and incubated with a secondary antibody (Dingguo, Guangzhou, China). The chromogenic reaction, counterstain, and dehydration were carried out. After that, the morphological changes in the brain microvascular tissue were observed under an ECLIPSE Ti2 optical microscope (with *Axio Imager M2* software v4.50) (Nikon, Tokyo, Japan).

### 2.4. The Rest Chemicals and Reagents

The chemicals and reagents used in this study were obtained as follows: Sigma-Aldrich (St. Louis, MO, USA) for 4′,6-diamidino-2-phenylindole (DAPI), nicotine (NT), and methyllycaconitine citrate (MLA); Thermo Fisher Scientific (Waltham, MA, USA) for fluorescent α-bungarotoxin conjugates; Proteintech (Proteintech Group, Chicago, IL, USA) for enzyme-linked immunosorbent assay (ELISA) kits; MedChemExpress (Princeton, NJ, USA) for Tryphostin AG 490 (AG490) and memantine (MEM). The rest of the reagents were purchased from Beyotime Institute of Biotechnology, Shanghai, China.

### 2.5. Statistical Analysis

Data are expressed as the mean ± standard deviation. SPSS 20.0 software (IBM, Almonk, USA) was used for statistical analysis. * *p* < 0.05, ** *p* < 0.01; **** p* < 0.000 were considered statistically significant.

## 3. Results

### 3.1. α7nAChR Functions as the Key Regulator in E. coli-Induced BBB Injuries In Vitro and In Vivo

We validated that a cholinergic antagonist (MLA) and agonist (nicotine, NT) affected α7nAChR binding activation in HBMECs, as detected by Alexa Fluor 488-conjugated alpha-bungarotoxin (alpha-BTX) (Figure 1B), suggesting that the fluorescence intensity of α-BTX decreased in MLA-pretreated cells compared to E44- and E44+NT-pretreated cells, and for the fluorescence intensity no significant differences were shown between E44 and E44+NT pretreated groups. In addition, adhesive and invasive results showed that *E. coli* E44 could productively infect HBMECs, while MLA attenuated the invasive abilities of host cells (*p* < 0.05) (Figure 1C,D) and NT increased E44 invasiveness capability but did not affect adhesion (*p* < 0.05) (Figure 1D). At the protein level, WB was applied to detect the expression of occludin, which is related to the integrity of tight junctions in HBMECs (Figure 1E). The results showed that occludin declined significantly in the E44 and E44+NT groups (*p* < 0.05), while MLA slightly improved *E. coli* E44-induced tight junction lesions (multiple comparisons). As known, an increase in proinflammatory factors (IL-6 and TNF-α levels) could enhance the permeability of the BBB, which is correlated with inflammation [20,21]. After 2 h of infection, the levels of IL-6 and TNF-α increased in the *E. coli* E44 group, when compared to the control group (Figure 1F,G). Pretreatment with MLA induced the downregulation of IL-6 and TNF-α levels (*p* < 0.01).


*“In vivo, α7nAChR-KO mice were used to verify the regulatory function of α7nAChR in the pathogenesis of BSM. PCR methods and agarose gel electrophoresis were applied to genetically identify 12 mice (wild type: n = 6; α7nACR-KO: n = 6). The size of the amplicon was observed at 390 bp for wild-type mice (number from wild type 1 to 6) and 187 bp for α7nAChR-KO mice (number from α7nAChR 1 to 6) (Figure 2B). In addition, mouse brain tissues were stained with α-bungarotoxin and DAPI and observed under an inverted fluorescence microscope (100×). α-BTX staining (red fluorescence) could be observed in wild-type mice compared to α7nAChR-KO mice, as detected by Alexa Fluor 555 conjugated α-bungarotoxin (Figure 2C). To determine the BSM, the bacterial load in blood and cerebrospinal fluid was detected by the dilution plate counting method. In the results, E. coli E44 counted in peripheral blood and cerebrospinal fluid consistently decreased in α7nAChR-KO mice when compared to wild-type mice (p < 0.05) (Figure 2D). On the other hand, brain tissues were collected for hematoxylin-eosin (H&E) staining and histological observation. Under the optical microscope (×200 and ×400 magnification), the typical morphology of the brain microvascular and meninges in the E44+WT group could be clearly observed, which was characterized by the collapse of endothelial cells and the extravasation of circulating RBCs into the tissue space (black arrow in Figure 2E). However, E44+α7nAChR-KO group exhibited a healthy BBB morphology, meaning that α7nAChR knockout attenuates BBB injuries induced by E44 infection (Figure 2E). From the above results in vitro and in vivo, it is reasonable to believe that α7nAChR could function as the key regulator in E. coli-induced BBB disruptions.”*


### 3.2. α7nAChR Triggers JAK2/STAT5 to Aggravate E. coli E44-Induced Injuries

Studies have suggested that JAK2-mediated signaling pathways are associated with impairments in endothelial cells and the BBB [22,23]. Therefore, IHC was applied to directly observe the expression of JAK2 in the brain microvascular (Figure 2F). In HBMECs, the expression of JAK2 in the WT group was higher, accompanied by damaged microvascular profiles, than that in α7nAChR-KO mice. These results indicated that JAK2 might be positively correlated with *E. coli* E44-induced BBB injuries. We also detected JAK2 expression in HBMECs infected with *E. coli* E44 with goat anti-rabbit IgG H&L (TRITC) by cell immunofluorescence. The results showed that red fluorescence intensity (JAK2) was enhanced in HBMECs infected with *E. coli* E44 when compared to the uninfected cells (Figure 3A).

In the next experiments, we explore the mechanism by which α7nAChR triggers JAK2/STAT5 using the JAK2 inhibitor AG490. Considering that AG490 has been reported to be cytotoxic [24], we determined the cytopathic effect of AG490 (different concentrations: 10, 50, 100 μM) on HBMECs. The results showed that the concentration (AG490: 50 μM) of treatment was appropriate for HBMECs (Figure 3B). HBMECs treated with AG490 50 μM and 10 μM did not show apparent cell death and morphological changes. Furthermore, *E. coli* E44 cultured in the presence of inhibitor AG490 exhibited viability, as determined in vitro. The results showed that *E. coli* E44 cultured in AG490:10, 50, and 100 μM exhibited identical growth curves with the untreated group during the culture period (Figure 3C), suggesting that AG490 has no direct influence on the growth of E44. Upon further study, we validated the protective effect of AG490 on *E. coli*-induced BBB injuries for the first time. The viability of E44 on HBMECs surface and intracellular of each group and IL-6 and TNF-α were used as indicators of invasion, adhesion, and inflammation after E44 infection. The results were as follows: (1) adhesive and invasive results showed that AG490 could productively prevent E. coli E44 infection into HBMECs (*p* < 0.05) (Figure 3D), and (2) IL-6 and TNF-α levels decreased in AG490-pretreated HBMECs supernatant (*p* < 0.05) (Figure 3E). To verify the mechanism of the intracellular JAK2–STAT5 signaling pathway in *E. coli*-induced BBB injuries, Western blotting was applied to detect the expression of JAK2, p-JAK2, STAT5b, p-STAT5b, and Occludin. The results showed that AG490 inhibited the activation of p-JAK2 and p-STAT5b and improved tight junctions (occludin) in infective HBMECs (*p* < 0.01) (Figure 3F). It is a meaningful finding that such a regulatory effect may be positively correlated with α7nAChR (*p* < 0.01) (Figure 3G). α7nAChR activation significantly upregulates the expression of JAK2 and STAT5b in HBMECs, and the involvement of the JAK2–STAT5 pathway in E44-induced BBB disruption has been proven (Figure 3F,G). As mentioned above, in the stage of BBB injuries induced by *E. coli* K1, α7nAChR is a key regulator and affects the integrity of HBMECs by mediating the JAK2–STAT5 signaling pathway, while AG490 inhibits the JAK2–STAT5 pathway and exhibits protective effects on *E. coli* K1 infection.

### 3.3. CISH Negatively Regulated JAK2/STAT5, Exhibiting Protective Effects during Infection

JAK/STAT activation is considered to be associated with the release of a variety of cytokines and proinflammatory factors. Recent studies found that CISH exhibits a relatively efficient and specific inhibitory effect on the JAK2–STAT5 pathway by binding to tyrosine residues phosphorylated cytokine receptor through its SH-2 domain and masking the docked site of STAT5. To explore whether CISH could inhibit the JAK2–STAT5 pathway during BSM, we used siRNA and overexpression techniques to silence/upregulate CISH in HBMECs, respectively. The results were as follows: compared with the untreated group, designed siRNA showed an acceptable inhibition effect of CISH with statistical significance (*p* < 0.05) (Figure 4B). The expression of Occludin in HBMECs treated with siRNA decreased significantly compared with that in HBMECs not treated with siRNA after infection (*p* < 0.01) (Figure 4C). Meanwhile, we also detected the level of the JAK2–STAT5 pathway in infective HBMECs pretreated with siRNAs, coupled with the upregulation of p-JAK2 and p-STAT5b in intracellular proteins by WB (*p* < 0.05) (Figure 4D). On the other hand, western blotting showed that CISH in HBMECs transfected with GV492-CISH increased by approximately 40% compared with that in the control group (*p* < 0.01), while there was no statistical difference between the NC group (no-load control plasmid) and the control group in terms of CISH expression (*p* > 0.05) (Figure 4E). In addition, the expression levels of p-JAK2 and p-STAT5b were decreased in HBMECs after GV492-CISH transfection, while tight junctions (occludin) were improved (*p* < 0.01) (Figure 4F,G). As mentioned above, CISH protein was proven to negatively regulate the JAK2–STAT5 pathway and improve the tight junction destruction of HBMECs upon *E. coli* E44 infection.

### 3.4. MLA and MEM Attenuated E. coli-Induced BBB Injuries Mediated by the α7nAChR-CISH Axis

To explore the protective effect of α7nAChR inhibitors MLA and MEM on E44 infection of HBMEC, in vitro, we applied the 2-[2-methoxy-4-nitrophenyl]-3-[4-nitrophenyl]-5-[2,4-disulfophenyl]-2*H*-tetrazolium (WST-8) assay, which is widely used to determine cell viability in the field of cell biology [25]. In the CCK-8 test, the absorbance at 450 nm was proportional to the number of living HBMECs in the culture (Figure 5A). The results verified the damaging effect of *E. coli* E44 on HBMECs (*p* < 0.05) and the improvements of MLA and MEM on BBB injuries (*p* < 0.05). In addition, ELISA results (proinflammatory cytokines IL-6 and TNF-α detected in cell supernatant) showed the downregulation of IL-6 and TNF-α in the MLA and MEM groups compared to the E44 infection group (*p* < 0.01) (Figure 5B). Morphological observations consistently confirmed the above viewpoints, pretreatment with MEM and MLA reduced the cell number reduction and morphological changes induced by E44 infection (Figure 5C). A preliminary study analyzed the transcriptome profile differences between E44-infected mouse brain microvascular endothelial cells (MBMEC) and E44+MEM MBMEC cells (Yu et al., 2015) and found that MEM could significantly upregulate anti-inflammatory factors (including CISH, etc.). Therefore, we detected the expression of CISH in infective HBMECs at the protein level by Western blotting (Figure 6) and cell immunofluorescence (Figure 5D). The results showed that MLA and MEM pretreatments could induce the upregulation of CISH in infective HBMECs (*p* < 0.01), which may function as an “executor” in inhibiting inflammatory signaling pathways during the pathogenesis of BSM. Our results showed that the expression of Occludin declined significantly in the *E. coli* E44 infection group compared with the uninfected group (*p* < 0.01). MLA and MEM induced an increase in CISH and exhibited a relatively good protective effect on infective HBMECs (*p* < 0.01) (Figure 5E) (*t*-test).

Thirty-two mice were randomly divided into four groups (control group, E44 group, MEM group, and MLA group). Mice in each group were given a normal saline solution containing MLA (5 mg/kg) or MEM (20 mg/kg) by intraperitoneal administration for 3 days. After *E. coli* E44 infection for 8–10 h, mouse brains were collected for morphological observation. H&E staining showed that *E. coli* E44 could induce impairments in endothelial cells and cause brain microvascular damage. Histological damage was characterized by microvascular collapse, the destruction of the bases, and RBC cells that were irregularly positioned into the tissue space. Few well-structured blood capillaries could be observed under a typical view (Figure 6A). Under a 400× amplification microscope, MEM-pretreated mice and MLA-pretreated mice exhibited relatively protective effects on *E. coli*-induced BBB injuries, as characterized by the more well-structured brain capillaries and morphological integrity of HBMECs. In addition, both MEM and MLA showed good improvements in *E. coli*-induced BSM, as determined by bacterial counts in blood and cerebrospinal fluid (*p* < 0.01) (Figure 6B,C). These results consistently suggested that MLA and MEM could attenuate *E. coli*-induced BBB injuries in vitro and in vivo.

### 3.5. α7nAChR/CISH/JAK2/STAT5 Axis Is Critical for E. coli-Induced BBB Disruptions

In this study, we found that *E. coli* E44 infection could trigger the α7nAChR/JAK2/STAT5 signaling pathway in HBMECs. Nicotine (α7nAChR agonist) aggravated *E. coli*-induced BBB injuries, while MLA and MEM (α7nAChR antagonist) exhibited protective effects by regulating the α7nAChR/CISH/JAK2/STAT5 signaling pathways during infection. CISH was proven to function as a negative executor, which inhibits the activation of p-JAK2 and p-STAT5b in infective HBMECs (Figure 7).

## 4. Discussion

Neonate and infant mortality rates are still important indicators for evaluating countries’ medical levels. It is worth confirming that antibiotics are effective in the management of pediatric BSM, but inappropriate antibiotic use may result in adverse effects and serious complications, including (1) the aggravation of an already fragile situation of bacterial drug resistance [26]; (2) increasing the risk of necrotizing enterocolitis (NEC) in late-onset sepsis due to imbalanced flora [5,6]; and (3) the lysis of killed bacteria triggering the production of pro-inflammatory factors and exacerbating the local inflammatory response [7,27]. For the reasons above, host-pathogen interactions have become a new strategy and research point for the development of antibiotic replacement therapies. In this study, bacterial loads in blood and cerebrospinal fluid were significantly reduced in α7nAChR-KO mice infected with *E. coli* E44 (Figure 2D), while pretreatment with the NT increased bacterial invasion (Figure 1D), indicating that α7nAChR mediates BSM caused by *E. coli* E44 infection. Our published studies have shown that α7nAChR functions as a key target in host-directed antimicrobial drugs against pathogens and plays an important role in the pathogenesis of *E. coli* BSM by enhancing *E. coli*-related formation and release of neutrophil extracellular traps (NETs) and suppressing polymorphonuclear neutrophils (PMNs) across the BBB in neuronal inflammation [10,11]. Therefore, it is meaningful for us to reveal the underlying mechanism of the in vivo nicotinamide metabolism-related receptor α7nAChR and its downstream signaling pathways in pediatric *E. coli* BSM.

In the study, the specific inhibitor molecules methyl lycaconitine citrate (MLA) and memantine hydrochloride (MEM) were used in BSM to explore the role and mechanism of α7nAChR. Specifically, in vitro, pretreatment with MLA (50 μM) or MEM (50 μM) significantly attenuated the survival rates of HBMECs and reduced the expression of inflammatory cytokines (such as IL-6 and TNF-α) after *E. coli* infection. In vivo, brain microvascular injuries and BBB lesions were improved in infected mice pretreated with MLA (5.0 mg/kg) or MEM (20.0 mg/kg), which is consistent with the pathological and immunohistochemical results. In addition, pretreatment with MLA and MEM significantly increased the expression of CISH, compared with the NT or *E. coli* E44 group in the immunofluorescence results of HBMEC (Figure 5D), indicating that inhibition of α7nAChR activation could increase the expression of CISH protein. As a drug approved by the FDA for long-term use in Alzheimer’s syndrome [21,28], MEM could also function as a mild antagonist of the *N*-methyl-d-aspartate receptor (NMDAR). Studies in vitro and in vivo demonstrated that MEM (IC50 = 0.34 μM) could more effectively attenuate meningitis injuries caused by Escherichia coli K1 strain than NMDA (IC50 = 5.1 μM) [29,30]. At the same concentration, there were no significant improvements in BSM when treated with NMDAR antagonists (DM), suggesting that NMDARs do not play a regulatory role during BSM infection [30]. On the other hand, MLA is considered a specific antagonist of α7nAChR, which has been proven to have certain efficacy in improving ulcerative colitis, Crohn’s disease, etc. [31]. Based on the results, both MEM and MLA exhibited the potential value of treatment for BSM. In this study, we successfully verified the protective effects of MLA and MEM on meningitic *Escherichia coli*-induced BBB disruptions.

Activation of the JAK/STAT signaling pathway is closely related to inflammatory injuries induced by infection [32,33]. In the study, we first validated that *E. coli* K1 infection triggers JAK2/STAT5 activation in HBMECs, while AG490 (JAK2 inhibitor) pretreatment and CISH overexpression inhibited JAK2–STAT5 and attenuated BSM injuries. Published studies suggest that the suppressor of cytokine signaling proteins (SOCS) are the main regulators in the innate immune reaction induced by a microbial pathogen, especially participating in the cellular negative feedback regulation of JAK/STAT [34], avoiding excessive inflammation [35]. Recent studies have found that SOCS proteins can function as executors and contribute to the downregulation of pro-inflammatory factors associated with infection [36,37]. As an important member of the SOCS family, the CISH protein has four binding sites targeting STAT5 and exhibits a higher specificity in suppressing the activation of the JAK2–STAT5 signaling pathway, eventually inhibiting the phosphorylation of STAT5 [38,39]. In this study, the CISH protein was first proven to be mediated by α7nAChR in HBMECs under bacterial infection. However, how α7nAChR precisely mediates CISH is still unknown. By putative transcription factor-binding sites (TFBSs) involving JASPAR and PROMO, we found that GTTGTTAACAAC of transcription factor FOXP3 covered the range in the CISH promoter region from 271 bp to 276 bp and 693 bp to 698 bp, with a dissimilarity margin of less than or equal to 0%. Published research reported that FOXP3 could be directly regulated by α7nAChR and positively correlated with the changes in CISH under infection conditions [40,41]. Hence, we speculated that α7nAChR–FOXP3 might act as a potential pathway to regulate CISH expression, which will be explored in the future.

## 5. Conclusions

In conclusion, the completion of this study well reveals the protective mechanism of CISH in α7nAChR-mediated meningitic *Escherichia coli*-induced HBMECs injuries and BBB disruptions. This study will provide the comprehensive pathogenesis mechanism of α7nAChR-mediated *E. coli* BSM and novel evidence for the development of α7nAChR antagonists in the prevention and treatment of pediatric *E. coli* BSM.

## Figures and Tables

**Figure 1 biomedicines-10-02358-f001:**
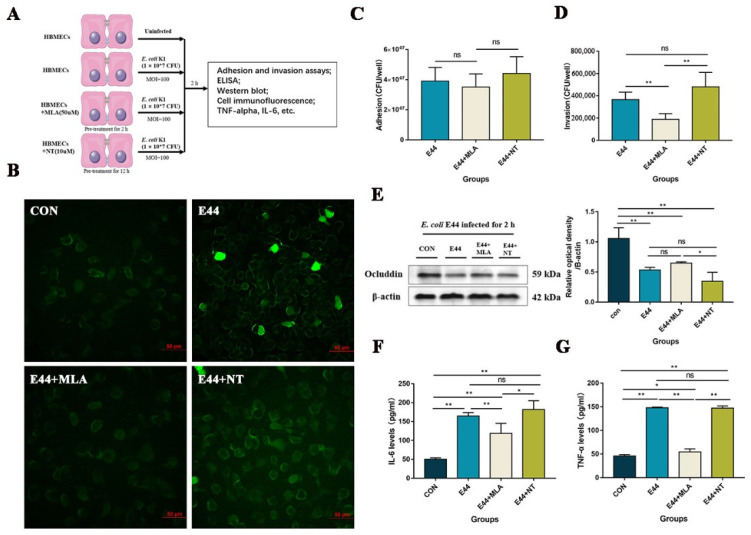
Effects of α7nAChR agonist (NT) and antagonist (MLA) on HBMECs infected with *E. coli* E44 in vitro. (**A**) Experiment design in vitro. (**B**) α7nAChR activation on pre-treated HBMECs infected with *E. coli* E44 detected by α-BTX assay, conducted by Axio Imager M2 software (Nikon, JPN). (**C**) Adhesive abilities of *E. coli* E44 infected to HBMECs pretreated with MLA and NT. (**D**) Invasive abilities of *E. coli* E44 infected to HBMECs cells pretreated with MLA and NT. (**E**) The expression of tight junction protein (Occludin) in HBMECs infected with *E. coli* E44 treated with MLA and NT, and relative optical density analyzed by Image J. (**F**) The expression of IL-6 in HBMEC cells’ supernatant with MLA and NT when infected with *E. coli* E44. (**G**) The expression of TNF-α in HBMEC cells’ supernatant. * *p* < 0.05, ** *p* < 0.01; ns, no significance.

**Figure 2 biomedicines-10-02358-f002:**
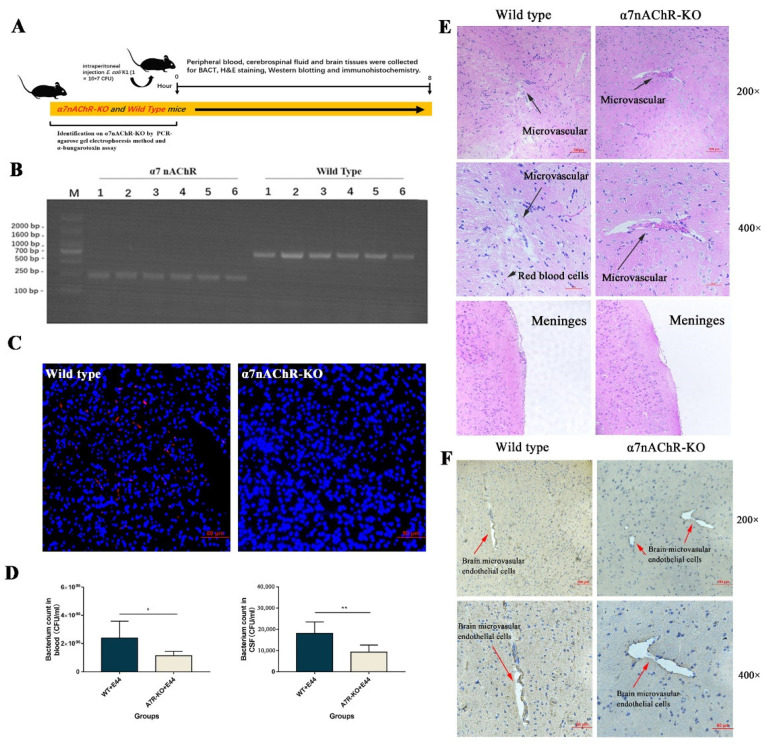
*E. coli* E44 induced BBB injuries in α7nAChR-KO and wild-type mice in vivo. (**A**) Experiment design in vivo. (**B**) Genotype of F2 generation α7nAChR knockout mice by PCR. (**C**) Fluorescence staining of α-Bungarotoxin in brain tissue collected from wild-type and α7nAChR-KO mice, conducted by Axio Imager M2 software (Nikon, JPN). (**D**) Bacterial load in blood and cerebrospinal fluid of WT/α7nAChR-KO mice infected with *E. coli* E44. (**E**) Morphological observations of brain microvascular and meninges in wild-type and α7nAChR-KO mice by H&E staining. (**F**) Morphological observations of brain microvascular in wild-type and α7nAChR-KO mice by JAK2 immunohistochemistry technique. * *p* < 0.05, ** *p* < 0.01.

**Figure 3 biomedicines-10-02358-f003:**
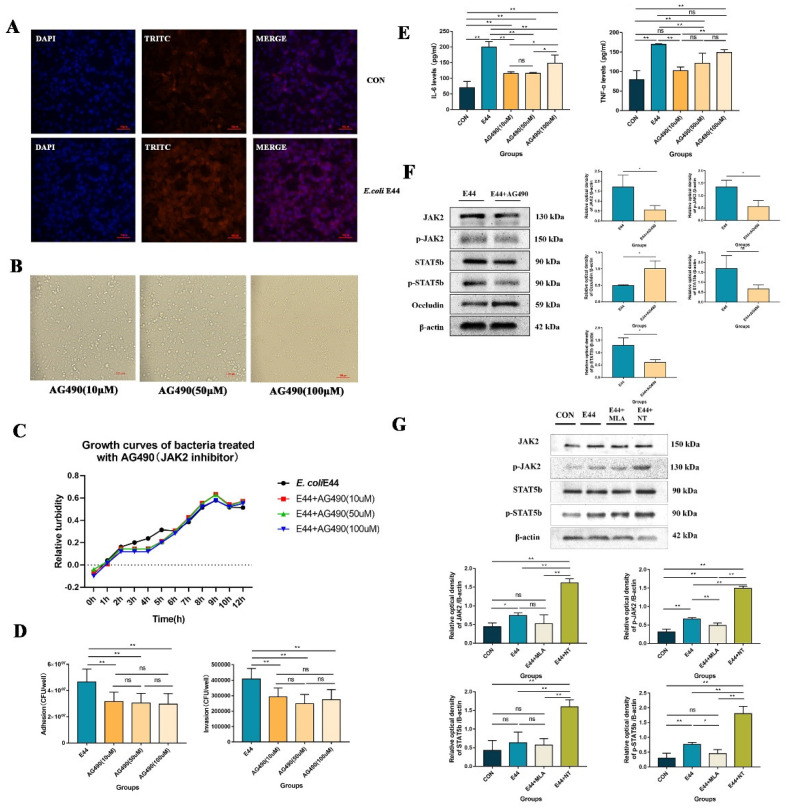
α7nAChR-mediated JAK2/STAT5 signaling pathways activation in HBMECs infected with *E. coli* E44. (**A**) The expression of JAK2 in HBMECs infected with/without *E. coli* E44 detected by cell immunofluorescence, conducted by Axio Imager M2 software (Nikon, JPN). (**B**) Morphological observations of HBMECs treated with different concentrations of AG490. (**C**) Antibacterial activities of AG490 on *E. coli* E44 were detected in vitro. (**D**) Adhesive, invasive abilities of *E. coli* E44 infected to HBMECs pretreated with different concentrations of AG490. (**E**) The expressions of IL-6 and TNF-α in HBMEC cells’ supernatant with AG490 when infected with *E. coli* E44. (**F**) The expressions of p-JAK2, p-STAT5b, and Occludin proteins in HBMECs with/without AG490 when infected with *E. coli* E44, and relative optical density analyzed by Image J. (**G**) The expressions of p-JAK2 and p-STAT5b in HBMECs with NT or MLA when infected with *E. coli* E44, and relative optical density analyzed by Image J. * *p* < 0.05, ** *p* < 0.01; ns, no significance.

**Figure 4 biomedicines-10-02358-f004:**
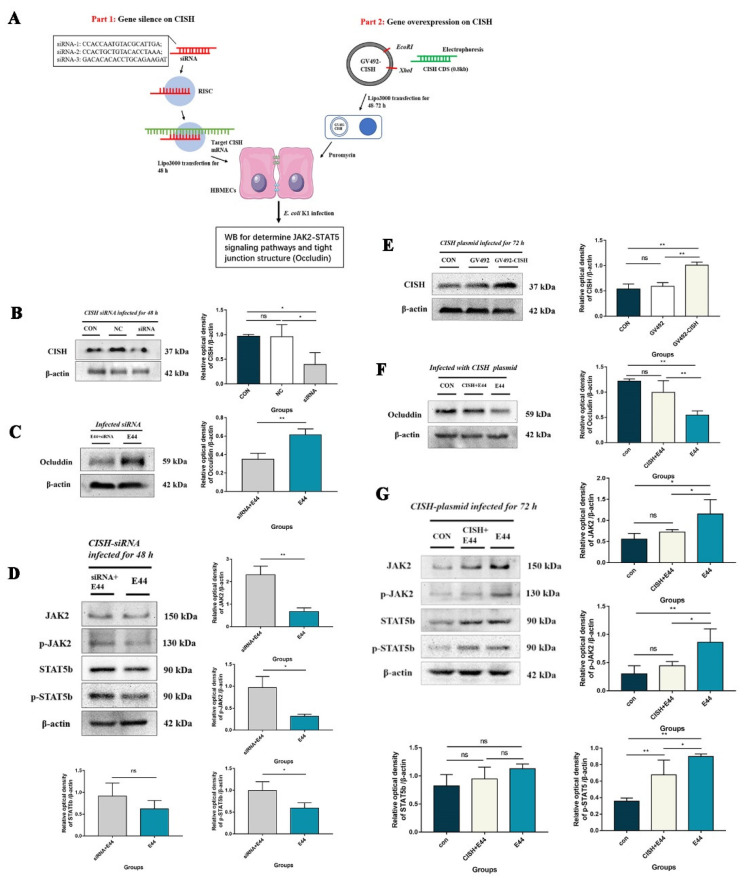
Gene silencing (siRNA) and over-expression (GV492) on the expression of CISH in HBMEC cells when infected with *E. coli* E44. (**A**) Experiment design in vitro. (**B**) The effect of siRNA-1, -2, and -3 on CISH expression in HBMECs, relative optical density analyzed by Image J. (**C**) The effect of siRNA on the tight junction protein (Occludin) in HBMECs treated with *E. coli* E44, relative optical density analyzed by Image J. (**D**) The expression of JAK2, p-JAK2, STAT5b and p-STAT5b in HBMEC cells (with/without) siRNA-1 when infected with *E. coli* E44, relative optical density analyzed by Image J. (**E**) The effect of GV492-CISH on CISH protein over-expressions in HBMEC cells. (**F**) The effect of GV492-CISH on the tight junction protein (Occludin) in HBMEC cells treated with *E. coli* E44. (**G**) The expression of JAK2, p-JAK2, STAT5b, and p-STAT5b in HBMEC cells (with/without) GV492-CISH when infected with *E. coli* E44, relative optical density analyzed by Image J. * *p* < 0.05, ** *p* < 0.01; ns, no significance.

**Figure 5 biomedicines-10-02358-f005:**
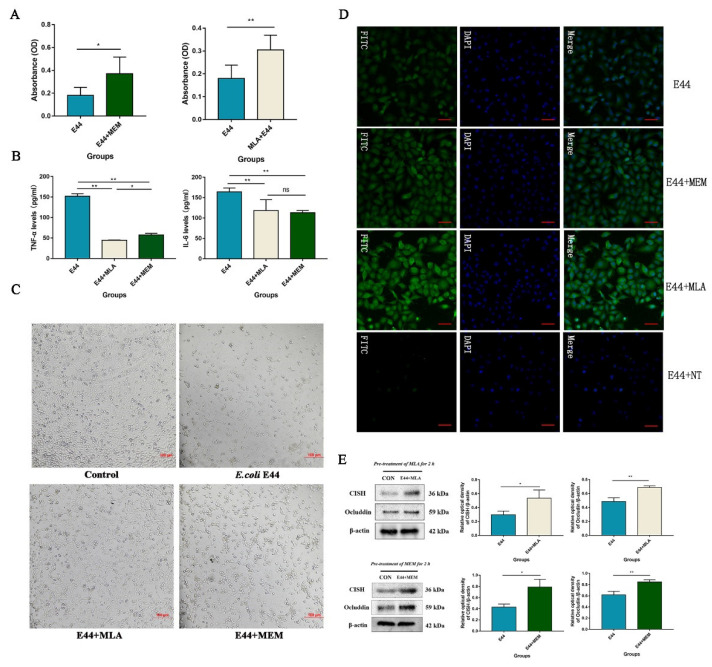
Protective effects and mechanism of MLA and MEM attenuating *E. coli*-induced HBMEC cell injuries in vitro. (**A**) Cell viabilities of infected HBMEC cells treated with MLA and MEM, detected by CCK-8 assay. (**B**) The expressions of IL-6 and TNF-α in HBMEC cells’ supernatant with MLA, MEM when infected with *E. coli* E44. (**C**) Morphological observations of infected HBMEC cells pre-treated with MLA and MEM, respectively (200×). (**D**) α7nAChR activation/inhibition on HBMEC treated with MLA, MEM, and NT, detected CISH expression by immunofluorescence assay, respectively, conducted by Axio Imager M2 software (Nikon, JPN). (**E**) The expressions of Occludin, CISH proteins in infected HBMECs treated with MLA, MEM. The expressions of CISH and Occludin in infected HBMEC cells pre-treated with MLA, MEM detected by WB, relative optical density analyzed by Image J. * *p* < 0.05, ** *p* < 0.01; ns, no significance.

**Figure 6 biomedicines-10-02358-f006:**
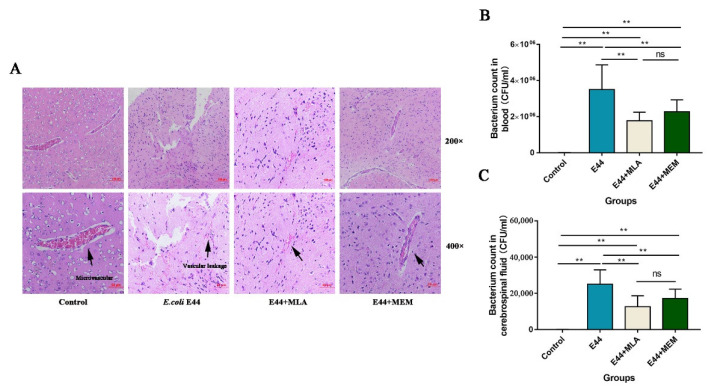
Protective effects of MLA and MEM attenuating BBB lesions and inflammation in *E. coli*-induced mice in vivo. (**A**) Morphological observations of brain microvascular in wild-type mice pretreated with MLA and MEM by H&E staining. (**B**) Bacteria counts in blood and cerebrospinal fluid of mice infected with *E. coli* E44 detected by the dilution method of plate counting. (**C**) Bacteria counts in cerebrospinal fluid of mice infected with *E. coli* E44 in different pretreat groups. ** *p* < 0.01; ns, no significance.

**Figure 7 biomedicines-10-02358-f007:**
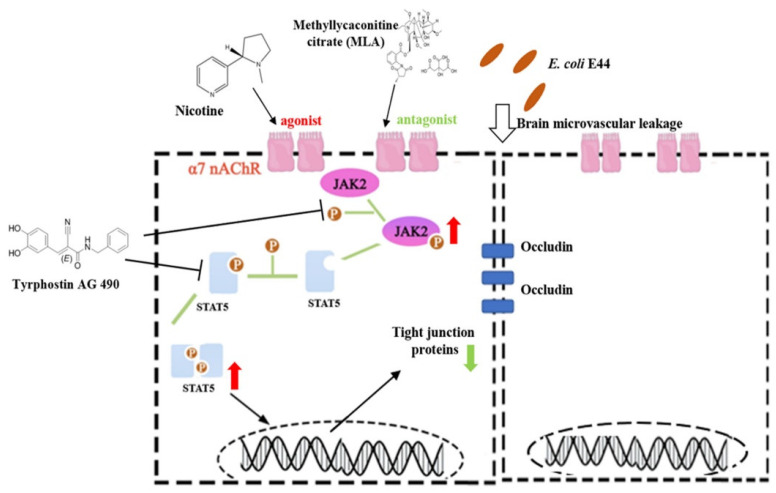
A systematic scheme of both α7nAChR/CISH/JAK2/STAT5 signaling pathways and inhibitors (AG490, MLA, and MEM) in the regulation of the pathogenesis of BBB disruptions.

## Data Availability

Not applicable.

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
