# Peer review of "Alpha7 Nicotinic Acetylcholine Receptor Antagonists Prevent Meningitic Escherichia coli-Induced Blood–Brain Barrier Disruptions by Targeting the CISH/JAK2/STAT5b Axis"

_biomedicines, 2022, doi:10.3390/biomedicines10102358_

Round 1

Reviewer 1 Report (Previous Reviewer 1)

While I appreciate the efforts made by the authors to address previous comments, several issues with the structure and presentation of this manuscript remain. Although the study is well designed and experimentally sound, the results section still needs major reformatting before the manuscript can be accepted for publication.

Minor comments:

Line 50: Please make the following correction: ‘’which are a significant indicator for evaluating…’’.

Line 53: It is mentioned that the alpha7nAChR plays a role in the pathogenesis of BSM and that it is associated with an increase in CISH expression. However, in this manuscript, a protective effect by CISH is demonstrated. This is in contradiction and should be discussed in the manuscript.

Line 61-63: Please make the following correction: ‘’in vitro ( ) and in vivo ( ) models, we evaluated….’’

Line 65-67: The last sentence does not make grammatical sense. Please correct.

Line 229: Please make following correction: ‘’In the next experiments, we explore the mechanism by which alpha7nAChR triggers….’’.

Line 234: Please make following correction: ‘’..E. coli cultured in the presence of inhibitor () exhibited…’’

Line 273: Please make following correction:’’….HBMECs upon E. coli E44 infection.’’

Line 276: A7R should be alpha7nAChR like everywhere else in the manuscript.

Line 390: 1)…….where is number 2)?

Line 451: ‘’the protective mechanism of alpha7nAChR…’’ How is the receptor protective? This sentence should be adjusted.

Major comments:

I find the results section extremely hard to follow especially section 3.2. Experiments should be presented in a logistic manner (panel 3F is currently introduced prior to 3D) and should be introduced with some context and rationale. Sections should be clearly structured with no going back and forth with different topics. Section 3.4 appears to be a repeat of section 3.1. Perhaps these two sections could be grouped.

Previous comment about Figure 1B hasn’t been addressed: Shouldn’t the agonist increase staining compared to infection alone? All I see in the presence of NT is more background. Based on this and results in panel C, NT does not appear to have any effect on infection. This should be discussed in the text. I did not see mention of this figure panel in the discussion as claimed by the authors in their response to reviewers.

Figure 2E: Please comment on the morphology of brain microvascular and meninges in KO animals.

Figure 2F: I would move this panel to the next section 3.2 in which the role of JAK/STAT is investigated.

Figure 3G should be described in more details.

Author Response

Thank you for your time and consideration! 

We have carefully read and discussed the reviewers’ comments. In order to respond the questions raised by reviewers, we made the following modifications to the manuscript.

Point 1: Line 50: Please make the following correction: ‘’which are a significant indicator for evaluating…’’.

Response 1:

Thank reviewer for the meticulous comments, which will greatly help improve our manuscript's accuracy.

As the reviewer suggested, this sentence has been rewritten (Line 51 in revised manuscript).

Point 2: Line 53: It is mentioned that the alpha7nAChR plays a role in the pathogenesis of BSM and that it is associated with an increase in CISH expression. However, in this manuscript, a protective effect by CISH is demonstrated. This is in contradiction and should be discussed in the manuscript.

Response 2:

We are sorry for the confusion.

Our preliminary studies demonstrated that α7nAchR played an important role in the pathogenesis of BSM, accompanied by increasing CISH at the transcriptome level (Ref. PMID: 25993608). In the study, the protective effect of CISH on E44-induced BBB injury was proven by inhibiting JAK-STAT signaling pathway, and these experimental results are consistent with the theme of the manuscript.

CISH are the main regulators in the innate immune reaction induced by a microbial pathogen, especially participating in cellular negative feedback regulation of JAK/STAT, which we also appropriately discuss in Line 469-472 in revised manuscript.

Point 3: Line 61-63: Please make the following correction: ‘’in vitro ( ) and in vivo ( ) models, we evaluated….’’

Response 3:

Thank reviewer for the meticulous comments. As the reviewer suggested, this sentence has been rewritten (Line 62-64 in revised manuscript).

Point 4: Line 65-67: The last sentence does not make grammatical sense. Please correct.

Response 4:

Thank reviewer for the valuable comments. This paragraph has been rewritten to provide a more approachable structure for manuscript (Line 66-68 in revised manuscript).

Point 5: Line 229: Please make following correction: ‘’In the next experiments, we explore the mechanism by which alpha7nAChR triggers….’’.

Response 5:

Thank reviewer for the valuable comments. As the reviewer suggested, this sentence has been rewritten (Line 255 in revised manuscript).

Point 6: Line 234: Please make following correction: ‘’..E. coli cultured in the presence of inhibitor () exhibited…’’

Response 6:

Thank reviewer for the valuable comments. As the reviewer suggested, this sentence has been rewritten (Line 261 in revised manuscript).

Point 7: Line 273: Please make following correction:’’….HBMECs upon E. coli E44 infection.’’

Response 7:

Thank reviewer for the valuable comments. As the reviewer suggested, this sentence has been rewritten (Line 305 in revised manuscript).

Point 8: Line 276: A7R should be alpha7nAChR like everywhere else in the manuscript.

Response 8:

Thank reviewer for the meticulous comments, which will greatly help improve the quality of our manuscript. We have uniformly used α7nAchR to represent A7R in the revised manuscript (Line 146,154,305 and 310 in revised manuscript).

Point 9: Line 390: 1)…….where is number 2)?

Response 9:

Thank you for the constructive comments. As the reviewer suggested, this sentence has been supplemented in order to provide a more approachable structure for manuscript (Line 429 in revised manuscript).

Point 10: Line 451: ‘’the protective mechanism of alpha7nAChR…’’ How is the receptor protective? This sentence should be adjusted.

Response 10:

As the reviewer suggested, the sentences have been rewritten, and corrected grammatical errors (Line 489 in revised manuscript).

Point 11: I find the results section extremely hard to follow especially section 3.2. Experiments should be presented in a logistic manner (panel 3F is currently introduced prior to 3D) and should be introduced with some context and rationale. Sections should be clearly structured with no going back and forth with different topics. Section 3.4 appears to be a repeat of section 3.1. Perhaps these two sections could be grouped.

Response 11:

Thank you for the constructive comments. As the reviewer suggested, section 3.2 has been rewritten in order to provide a more reasonable and approachable structure for the manuscript.

In the revised manuscript, we exchanged the positions of Figure.3F and Figure.3E to improve the logic of the manuscript results. Figure.3B and Figure.3C are the cytotoxicity assay results of AG490-pretreated HBMECs. Figure.3D shows the protective effect of AG490 on the invasion and adhesion of E44 and updated Figure.3E shows the levels of inflammatory factors IL-6 and TNF-α in each group. For invasive adhesion assay and IL-6 and TNF-α detection, we added the appropriate background and rationale in section 3.2 (Line 267 in revised manuscript).

Given the similar content of section 3.4 and section 3.1, our explanation is as follows: in section 3.1, we explored the mechanism of specific activation or inhibition of α7nAchR in E44-induced BBB damage, focusing on the role of α7nAchR in infection. Section 3.4 further investigated the protective role of two α7nAchR antagonists (MLA and MEM) in BBB injury induced by E44 infection. Highlight the effects of two small-molecule compounds MLA and MEM, which could serve as an alternative to antibiotics for patient treatment in the future. Therefore, we considered it necessary to separate sections 3.1 and section 3.4 to highlight the points of each section.

Point 12: Previous comment about Figure 1B hasn’t been addressed: Shouldn’t the agonist increase staining compared to infection alone? All I see in the presence of NT is more background. Based on this and results in panel C, NT does not appear to have any effect on infection. This should be discussed in the text. I did not see mention of this figure panel in the discussion as claimed by the authors in their response to reviewers.

Response 12:

Thank you very much for the constructive suggestion and recommendation.

In Figure 1B, the fluorescence intensity no significant differences were shown between E44 and E44+NT pretreated groups. We added an explanation of the NT fluorescence results in section 3.1 and supplemented the results of the invasion and adhesion assay of NT-pretreated HBMECs infected with E44 to improve the structure and logic of the article (Line 204-208 in revised manuscript).

Thanks again for your valuable comments.

Point 13: Figure 2E: Please comment on the morphology of brain microvascular and meninges in KO animals.

Response 13:

Thank you for the constructive comments.

We have added a morphological description of brain microvascular and meninges of α7nAchR-KO animals in Figure.2E in section 3.1 (Line 233-235 in revised manuscript).

Point 14: Figure 2F: I would move this panel to the next section 3.2 in which the role of JAK/STAT is investigated.

Response 14:

Thank you for your constructive comments, which will be of great help in improving the quality of our manuscript.

We have moved the results from Figure.2F to section 3.2 to explain the role of JAK2-STAT pathway in E44 infection. Thank you for your constructive suggestion and recommendation, which allows the results of our experiments to be presented to readers with a reasonable and approachable structure (Line 246-250 in revised manuscript).

Point 15: Figure 3G should be described in more details.

Response 15:

As the reviewer suggested, section 3.2 added details of Figure.3G to provide a more reasonable and more approachable structure for manuscript (Line 279-281 in revised manuscript).

Reviewer 2 Report (New Reviewer)

The manuscript by Gong et al.  with a title“Alpha7 nicotinic acetylcholine receptor antagonists prevent meningitic Escherichia coli-induced blood brain barrier disruption by targeting CISH/JAK2/STAT5b axis” is an interesting work with presenting the results with a clinical relevance for the treatment of pediatric E. coli bacterial sepsis and meningitis (BSM). The conclusion is based on the results and the results are based on in vitro and in vivo experiments. Of particular interest are the effects of two small- molecule compounds MLA and MEM, which could serve as an alternative to antibiotics for patient treatment in the future. The present study is a continuation of the authors’ previous study of two publications, demonstrating experience on the subject and an expert knowledge. The present study benefits from this.

The abstract is concise and contains all the necessary information. The introduction is informative and interesting. The scientific purpose and a challenging unmet need for the study are well elaborated. The material and methods section is written with sufficient detail, supplemented with citations of the authors’ previous work. The description of most of the methods allows the reproduction of experiments.

Comments:

Please explain „MLA“ and „MEM“ abbreviations in the abstract

AG490 is not mentioned in either “Material and Methods” section nor in the abstract

Nicotine is not mentioned in the “Material and Methods”

The font size in the figures should be increased as most of the text in figures is not easy to read

The phase contrast images in Figure 3 are of poor quality and should be improved by e.g. adjusting the contrast to make the cells more visible

Author Response

Thank you for your time and consideration! 

We have carefully read and discussed the reviewers’ comments. In order to respond the questions raised by reviewers, we made the following modifications to the manuscript.

Point 1: Please explain „MLA“ and „MEM“ abbreviations in the abstract

Response 1:

Thank reviewer for the meticulous comments, which will be of great help to improve the accuracy of our manuscript. In the abstract, we have added the full names of MEM (Memantine hydrochloride) and MLA (Methyllycaconitine citrate) to make it easy for readers understand (Line 23 in revised manuscript).

Point 2: AG490 is not mentioned in either “Material and Methods” section nor in the abstract

Response 2:

Thank you for your constructive comments, which will be of great help in improving the quality of our manuscript.

We have added a new section 2.4 "The Rest Chemicals and Reagents" in the manuscript, introducing additional drugs and Reagents used in this study, including MEM, MLA, AG490, and NT. In the abstract, no mention was made of AG490-related experimental results to highlight the research topic and make the paragraphs logical

Thanks again for your valuable comments.

Point 3: Nicotine is not mentioned in the “Material and Methods”

Response 3:

Thank reviewer for the valuable comments, which will greatly help to improve the quality of our manuscript.

Nicotine-related information has been supplemented in a new section 2.4 "The Rest Chemicals and Reagents" in manuscript (Line 183-190 in revised manuscript).

Point 4: The font size in the figures should be increased as most of the text in figures is not easy to read

Response 4:

We are sorry for the confusion.

We have enlarged the font size in Figure 2 and Figure 6 in revised manuscript to make it easy for the reader to read (Line 367 and Line 414 in revised manuscript).

The figures in the manuscript were of sufficient clarity and the font size in the figures was not considered to affect the reading. If you need a clearer figure please contact the editor.

Thanks again for your valuable comments.

Point 5: The phase contrast images in Figure 3 are of poor quality and should be improved by e.g. adjusting the contrast to make the cells more visible

Response 5:

Thank reviewer for the valuable comments, which will greatly help improve the quality of our manuscript.

We adjusted the contrast of Figure 3 to make the cells more visible (Line 377 in revised manuscript).

Round 2

Reviewer 2 Report (New Reviewer)

The authors have responded to all my comments and have made the respective changes in the manuscript. I have no more suggestions.

This manuscript is a resubmission of an earlier submission. The following is a list of the peer review reports and author responses from that submission.

Round 1

Reviewer 1 Report

In this manuscript, Gong and colleagues investigate the underlying mechanisms of the involvement of Alpha7nAChR in the pathogenesis of BSM. They demonstrate that the receptor is responsible for the activation of the JAK2-STAT5 pathway upon infection leading to inflammation and disruption of the BBB which could be ameliorated by two receptor antagonists. This study has implications for the development of alpha7nAChR inhibitors as treatments against BSM.

Major modifications to the structure of this manuscript are required to ease the reading. Several results lack explanation and are not properly introduced. Controls are also missing in some experiments. I therefore reject this manuscript in its current state.

Minor comments:

Line 35: ‘’these deaths’’ refers to which deaths since none were mentioned until that point in the manuscript? This should be rephrased.

Line 50: ‘’the typical pathological features for evaluating BBB integrity.’’ This sentence is not clear and should be rephrased.

Line 56-56: ‘’JAK could be generated by aggregation and phosphorylation by alpha7nAChR’’. Please explain or rephrase. How is a protein generated by aggregation?

Line 61-63: ‘’in vitro ( ) and in vivo ( ) models, we dissected….’’

Line 67: A summary of findings with the two inhibitors should be provided in this paragraph rather than just a mention that it will be discussed.

Line 197: ‘’MLA may induce the downregulation of IL-6 and TNF-a levels’’ I don’t know what that means ‘’may induce’’. It either does or it does not.

Line 198-200: Take away instructions from the final text.

Line 234: ‘’..E. coli cultured in () exhibited…’’ cultured in what? There is clearly a word missing in this sentence. I think it is cultured in the presence of the inhibitor.

Line 240: ‘’…prevent E. coli E44 infection into HBMECs…’’

Line 401: The fact that this study is meaningful for the authors does not justify it. A more appropriate reasoning of why this should be investigated should be included.

Line 405-406: This sentence does not make grammatical sense. Please rephrase for clarity.

Line 450-452: Take away instructions from the final text.

Claudin 5 (amongst others) is a well-known dominant component of the BBB. Its expression upon infection and treatment should be determined in addition to occludin to confirm BBB integrity.

Major comments:

I find the results section extremely hard to follow. Experiments should be presented in a logistic manner and should be introduced with some context and rationale. Sections should be clearly structured with no going back and forth with different topics. For example at line 230, the use of the AG490 inhibitor appears completely random with no introduction of why it was included at this point.

Figure 1B: Shouldn’t the agonist increase staining compared to infection alone? All I see in the presence of NT is more background. Based on this and results in panel C, NT does not appear to have any effect on infection. This should be discussed in the text.

Figure 2B: More details should be provided to explain why these amplicon sizes were expected and what they represent. This could go in the methods section.

Figure 2C: Are the authors claiming that no infection is observed in KO mice and therefore that the alpha7nAChR is essential for infection? My understanding is that this protein contributes to pathogenesis but is not required for infection. This should be discussed. The measurement of JAK2 is random in this figure. It should be discussed why the authors decided to look at this protein at this point in the study.

Figure 3B: The images should be better explained. What am I looking at and why are the authors claiming no effect?

Figure 3F-3G: There is no mention of panel 3F in the text. Figure 3G should be discussed in more details. What are the authors trying to demonstrate here and what are their conclusions?

Figure 4: What is the NC sample and why was it include in certain panels only and not all or none? The difference in expression of JAK2 and pJAK2 is very hard to see by WB in panel D while the quantification is claimed significant. Perhaps a more convincing image should be included. Panel E is not necessary as part of the main manuscript. This could go in the supplementary material.

Figure 5A: There should be a non-treated control here to compare the effect of infection. MLA and MEM should be introduced before jumping into the results. What are they and why were they included here? What are the authors trying to demonstrate with these inhibitors?

Figure 5B: How does one measure concentration (presented on the Y axis) by ELISA?

Figure 5C: This panel needs to be explained with more details. What am I looking at and what are the main observations?

Figure 5D: This panel should be discussed with more details. I don’t see any difference in terms of fluorescence between the conditions.

Figure 6A: The results should be discussed in the text. Should I see differences between the presented brains and which ones? What are the main observations and what does it mean? If not, why is this illustrated here?

I would also recommend reviewing the entire manuscript to address several grammatical mistakes.

The discussion is more focused on previous studies than the current one. The results from this study should be discussed in more details with a focus on their meanings and applications.

Reviewer 2 Report

The submitted manuscript describes studies on the role of α7nAChR antagonist to prevent BBB disruptions induce by E. coli K1. They found that α7nAChR functioned as the key regulator that affects the integrity of HBMECs by activating the JAK2-STAT5 signaling pathway, while CISH inhibited JAK2-STAT5 activation and exhibited protective effects against E. coli infection.

The reviewer have serious concerns about the design of the study, in particular about the in vitro model used and cannot accept the manuscript for publication in Biomedicines before the authors clarify several points (not exhaustive):

(i) The authors should explain how they have obtained the cells. They declare that “HBMECs were obtained from ScienCell Research Laboratories (USA)” but the web site of this company says that the product is only available for purchase for domestic customers. https://www.sciencellonline.com/products-services/primary-cells/human/cell-types/endothelial-cells/human-brain-microvascular-endothelial-cells.html

(ii) In addition the authors stated that "according to the cell specifications, HBMECs are appropriate for studying the molecular and cellular properties of the BBB".

The authors should bring evidence that the cells display these properties at least by showing the data they obtained to the reviewers, data that will not be shown finally in the manuscript. Are the properties of the cells innate or induced ? How are they induced ?

(iii) The authors should rewrite the conclusion. Currently it does not sound as a conclusion, but rather as a sentence explaining how the authors could complete the conclusion:“Authors should discuss the results and how they can be interpreted from the perspective of previous studies and of the working hypotheses. The findings and their implications should be discussed in the broadest context possible. Future research directions may also be highlighted»